**Data Availability Statement:** The relevant minimal data set is within the Supporting Information files.

**Funding:** The author(s) received no specific funding for this work.

# Organizing pneumonia of COVID-19: Time-dependent evolution and outcome in CT findings

**Yan Wang**[1☯], **Chao Jin**[1☯], **Carol C. Wu**[2], **Huifang Zhao**[1], **Ting Liang**[1], **Zhe Liu**[1], **Zhijie Jian**[1], **Runqing Li**[1], **Zekun Wang**[3], **Fen Li**[3], **Jie Zhou**[4], **Shubo Cai**[4], **Yang Liu**[5], **Hao Li**[6], **Yukun Liang**[7], **Cong Tian**[1], **Jian Yang**[ID][1]*

**1** Department of Radiology, The First Affiliated Hospital of Xi'an Jiaotong University, Xi'an, P.R. China, **2** Department of Thoracic Imaging, University of Texas M.D. Anderson Cancer Center, Houston, TX, United States of America, **3** Department of Radiology, The Eighth Hospital of Xi'an, Xi'an, P.R. China, **4** Department of Radiology, Xi'an Chest Hospital, Xi'an, P.R. China, **5** Department of Cardiology, The First Affiliated Hospital of Xi'an Jiaotong University, Xi'an, P.R. China, **6** Department of Critical Care Medicine, The First Affiliated Hospital of Xi'an Jiaotong University, Xi'an, P.R. China, **7** Department of Radiology, Ankang Center Hospital, Ankang, P.R. China

☯ These authors contributed equally to this work.
* yj1118@mail.xjtu.edu.cn

## Abstract

### Background

As a pandemic, a most-common pattern resembled organizing pneumonia (OP) has been identified by CT findings in novel coronavirus disease (COVID-19). We aimed to delineate the evolution of CT findings and outcome in OP of COVID-19.

### Materials and methods

106 COVID-19 patients with OP based on CT findings were retrospectively included and categorized into non-severe (mild/common) and severe (severe/critical) groups. CT features including lobar distribution, presence of ground glass opacities (GGO), consolidation, linear opacities and total severity CT score were evaluated at three time intervals from symptom-onset to CT scan (day 0–7, day 8–14, day > 14). Discharge or adverse outcome (admission to ICU or death), and pulmonary sequelae (complete absorption or lesion residuals) on CT after discharge were analyzed based on the CT features at different time interval.

### Results

79 (74.5%) patients were non-severe and 103 (97.2%) were discharged at median day 25 (range, day 8–50) after symptom-onset. Of 67 patients with revisit CT at 2–4 weeks after discharge, 20 (29.9%) had complete absorption of lesions at median day 38 (range, day 30–53) after symptom-onset. Significant differences between complete absorption and residuals groups were found in percentages of consolidation (1.5% vs. 13.8%, $P = 0.010$), number of involved lobe > 3 (40.0% vs. 72.5%, $P = 0.030$), CT score > 4 (20.0% vs. 65.0%, $P = 0.010$) at day 8–14.

**Competing interests:** The authors have declared that no competing interests exist.

## Conclusion

Most OP cases had good prognosis. Approximately one-third of cases had complete absorption of lesions during 1–2 months after symptom-onset while those with increased frequency of consolidation, number of involved lobe > 3, and CT score > 4 at week 2 after symptom-onset may indicate lesion residuals on CT.

## Introduction

Since late December 2019, the ongoing outbreak of Coronavirus Disease 2019 (COVID-19) related pneumonia, caused by a novel coronavirus, severe acute respiratory syndrome corona-virus 2 (SARS-CoV-2; previously known as 2019-nCoV), has rapidly expanded throughout worldwide [1–3]. By 29 April 2020, a total of 3.01 million patients with confirmed COVID-19 pneumonia and 207,973 deaths have been reported [4]. Clinical and radiological characteristics of COVID-19 pneumonia have been systematically described. It is noting that the most common findings on chest computed tomography (CT), i.e. peripheral ground glass opacity (GGO), consolidation or both predominantly in bilateral and multifocal distributions highly resembled to a CT pattern of organizing pneumonia (OP) [5, 6]. As a common lung injury, most cases of OP were demonstrated to have a good prognosis, while permanent damage and interstitial fibrosis were still observed in scare severe cases [7]. Similar prognosis was observed in COVID-19, i.e. above 80% of cases had been discharged with recovery [8]. Despite this, prognosis of OP pattern in COVID-19 including radiological outcome and disease course relating to resolution of pulmonary lesions remains currently unclear.

A plenty of studies have explored the evolution of pulmonary lesions based on chest CT [9, 10]. As the disease progresses, increased number, extent and density of GGOs on CT have been observed [11]. Among these, consolidation was considered as an indication of poor prognosis [12]. However, evolutions of OP pattern in COVID-19 and the relations with radiological outcome have not been well described. This study therefore aimed to delineate the time-dependent evolution of CT findings and outcome in COVID-19 patients with OP pattern.

## Materials and methods

### Patients

This multicenter retrospective study was launched by the First Affiliated Hospital of Xi'an Jiaotong University, and approved by all the multicenter institutions, including the First Affiliated Hospital of Xi'an Jiaotong University, the Eighth Hospital of Xi'an, Xi'an Chest Hospital, Ankang Center Hospital, Wuhan No.9 Hospital, Hanzhong Center Hospital, Baoji Center Hospital. For ethic issue, IRB protocol of XJTU1AF2020LSK-011 is multiple-center. Informed patients' consent was waived with approval. 158 laboratory-confirmed patients with COVID-19 pneumonia who underwent chest CT scans between 22 January 2020 and 16 March 2020 were collected from seven hospitals in China. Among the patients, 75 were from Xi'an region; 18 were from Ankang region; 17 were from Hanzhong region; 10 patients were from Baoji region, and 38 patients were from Wuhan region. A case of COVID-19 was confirmed by a positive result on next-generation sequencing or real-time RT-PCR. The pulmonary lesions were considered to belong to OP pattern based on baseline CT: (1) peripheral predominantly GGO, consolidation or both, with subpleural or bronchovascular bundles distribution; (2) lobar involvement characterized by the total CT score less than or equal to 10 (Evaluation for

total CT score detailed below). The pulmonary lesions were considered to unmatched with OP pattern: (1) extensive GGO and/or consolidation diffusely in both lungs; (2) total CT score greater than 10; (3) combined consolidation and GGO with bronchial or bronchiolar wall thickening [13]. Patients which were with unmatched with OP pattern or unqualified CT images were excluded.

According to clinical classification from preliminary diagnosis and treatment protocols for novel coronavirus pneumonia (7th edition) of the National Health Commission, China [14], all patients were assessed as mild, common, severe and critical types and categorized into non-severe (mild/common) and severe (severe/critical) groups. For mild patients, clinical symptoms are subtle, and no pneumonia found on chest imaging. Patients with common type show symptoms such as fever and respiratory tract, and lung opacities on chest imaging. Patients with severe type should meet any of the following conditions: (1) respiratory distress, $RR \geq 30$ beats / min; (2) resting blood oxygen saturation $\leq 93\%$; or (3) partial pressure of arterial blood oxygen (PaO2)/oxygen concentration (FiO2) $\leq 300$ mmHg. Critical patients should meet one of the following criteria: (1) respiratory failure with mechanical ventilation; (2) shock; (3) other organ failure needing intensive care unit (ICU) treatment.

All the patients were diagnosed, isolated and hospitalized, which included initiation of anti-virals, interferon, Chinese herbal medications, supplemental oxygen [14]. The discharge criteria were: (1) body temperature returned to normal for greater than 3 days; (2) respiratory symptoms significantly improved; (3) pulmonary imaging showed obvious improvement in acute exudative lesions; (4) two consecutive negative COVID-19 nucleic acid tests at least 24 h apart. The discharged patients were recommended to quarantine for two weeks and then revisit the hospital with nucleic acid test and chest CT scan at 2 and 4 weeks after discharge [14]. The pulmonary sequelae, i.e. complete absorption or residuals with linear opacities, and/ or a few GGO with/without consolidation on revisit CT were evaluated. The disease course was defined as the interval from symptom onset to discharge.

## CT image acquisition

Chest CT scans were performed in 16- to 64-multidector CT scanners (Philips Brilliant 16, Philips Healthcare; GE LightSpeed 16, GE Healthcare; GE VCT LightSpeed 64, GE Healthcare; Somatom Sensation 64, Siemens Healthcare; Somatom AS, Siemens Healthcare; Somatom Spirit, Siemens Healthcare; GE Optima 680, GE Healthcare). The CT parameters included: 120 kVp of tube voltage, current intelligent control (auto mA) of 30–300 mA, and slice thickness/ slice interval of 0.6–1.5 mm.

## Image data collection and evaluation

Two experienced radiologists with respective 5 and 10 years of thoracic imaging experiences reviewed and described CT findings according to a peer-reviewed literature of viral pneumonia [9, 11]. The following terms were used: pure GGO; pure consolidation; GGO and consolidation; pure linear opacity; GGO and linear opacity; consolidation and linear opacity; GGO, consolidation and linear opacity; crazy paving pattern; reversed halo pattern. The pulmonary abnormalities involvement was quantitatively estimated by a semi-quantitative scoring system [15]. Each of the five lung lobes was visually scored from 0 to 4 as: 0, no involvement; 1, < 25% involvement; 2, 25%-49% involvement; 3, 50%-75% involvement; 4, > 75% involvement. The sum of the individual lobar scores were the total CT scores, which ranged from 0 (no involvement) to 20 (maximum involvement).

CT findings were designated to three time groups based on the time from symptom onset to CT scan (day 0–7, day 8–14, day > 14).

## Statistical analysis

Continuous variables were presented as mean ± standard deviation and the categorical variables were presented as the number and percentage of the total. Differences of CT findings across time groups (day 0–7, day 8–14, day > 14), between non-severe and severe cases, between complete absorption and residuals groups were analyzed by Chi-square test, Fisher's exact test, dependent sample t-test or Mann-Whitney U test as appropriate. Multiple comparisons were corrected by Bonferroni correction. Continuous variable on CT images, i.e. total CT score with significant difference in two-group comparison was further treated as categorical variable using an optimal threshold to maximize the Youden index of the receiver operating characteristic (ROC) analysis in discrimination of complete absorption vs. residuals groups.

All the statistical analyses were performed in the IBM SPSS Statistics Software (version 22; IBM, New York, USA). $P < 0.05$ was considered statistically significant.

## Results

### Patient characteristics

Of 158 patients, 106 (67.1%) were OP pattern, 3 (1.9%) were with negative CT, and 49 (31%) were unmatched with OP pattern (Fig 1). Of 106 OP pattern cases, 79 (74.5%) were non-severe and 27 (25.5%) were severe; 61 (61.6%) were with elevated C-reactive protein and 42 (40.0%) were with decreased lymphatic percentage (Table 1). The mean age was 48.0 ± 15.4 years and showed no significant gender difference (male, 45 [43.7%]; female, 61 [56.3%]). No patient had co-infection in our cohort.

Of 106 OP pattern cases, 103 (97.2%) were discharged, 2 (1.9%) were admitted to ICU, 1 (0.9%) died. Median times from symptom onset to discharge, to ICU admission, to death were 25 (range, 8–50) days, 24 days, 28 days, respectively. Of 67 patients with revisit CT after

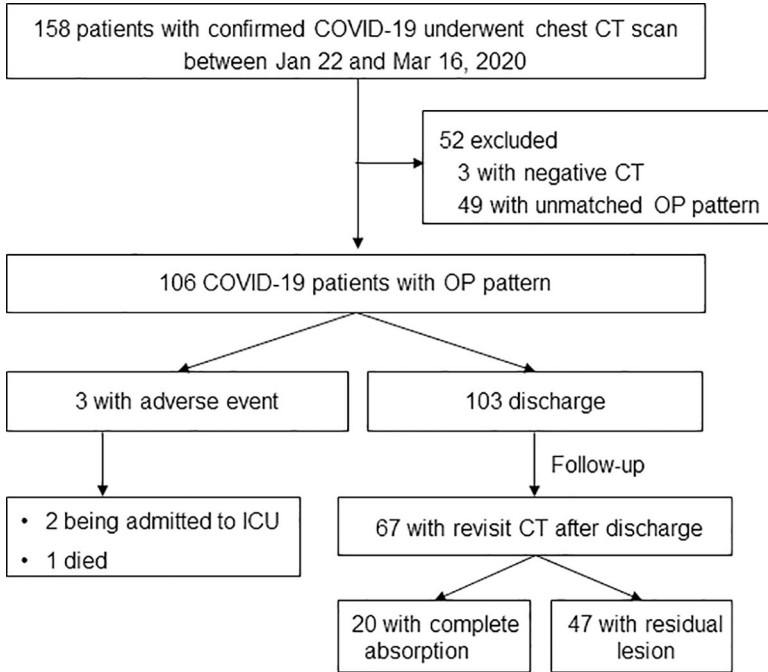

**Fig 1. Patients recruitment flowchart.** COVID-19 = coronavirus disease 2019, ICU = intensive care unit, OP = organizing pneumonia.

**Table 1. Demographics, laboratory test and clinical outcome in COVID-19 patients with organizing pneumonia pattern.**

| Characteristics | Patients |
|---|---|
| | (*n* = 106) |
| **Age (year)[a]** | 48.0 ± 15.4 |
| **Male sex** | 45 (43.7%) |
| **Disease severity** | |
| Non-severe | 79 (74.5%) |
| Severe | 27 (25.5%) |
| **Comorbidity** | 38 (35.8%) |
| **Exposure history** | |
| Recent travel to Wuhan | 63 (61.2%) |
| Contact with infected patient | 28 (27.2%) |
| Unknown exposure | 12 (11.7%) |
| **Initial symptom** | |
| Fever | 91 (88.4%) |
| Cough | 61 (59.2%) |
| Expectoration | 21 (20.4%) |
| Fatigue | 13 (12.6%) |
| Chest tightness and/or breath shortness | 21(20.4%) |
| Pharyngalgia | 10 (9.7%) |
| Muscle soreness | 7 (6.8%) |
| Headache | 7 (6.8%) |
| Nausea and/or vomiting | 1 (1.0%) |
| Diarrhea | 4 (3.9%) |
| No obvious symptoms | 2 (1.9%) |
| **Laboratory test at admission[b]** | |
| C-reactive protein (-,↑,↓) | 38 (38.4%), 61 (61.6%), 0 (0) |
| Percentage of lymphocytes (-,↑,↓) | 60 (57.1%), 3 (2.9%), 42 (40.0%) |
| Lymphocyte count (-,↑,↓) | 64 (61.5%), 0 (0), 40 (38.5%) |
| Percentage of monocytes (-,↑,↓) | 75 (72.1%), 26 (25.0%), 3 (2.9%) |
| White blood cell count (-,↑,↓) | 73 (69.5%), 2 (1.9%), 2 (28.6%) |
| Alanine Aminotransferase (-,↑,↓) | 82 (78.8%), 20 (19.2%), 2 (1.9) |
| Aspartate Aminotransferase (-,↑,↓) | 82 (78.8%), 21 (20.2%), 1 (1.0%) |
| Creatine kinase (-,↑,↓) | 84 (83.2%), 7 (6.9%), 10 (9.9%) |
| Neutrophilic granulocyte percentage (-,↑,↓) | 69 (65.7%), 26 (24.8%), 10 (9.5%) |
| Hemoglobin (-,↑,↓) | 79 (76.0%), 6 (5.8%), 19 (18.3) |
| **Clinical outcome** | |
| Discharge | 103 (97.2%) |
| Admission to ICU | 2 (1.9%) |
| Death | 1 (0.9%) |

Note: Unless otherwise indicated, data are reported as the number of patients, with percentages in parentheses.

[a], data were reported as the mean ± standard derivation.

[b], -, ↑, ↓ represent within, above, and below normal ranges of laboratory results, respectively. Normal ranges of C-reactive protein, percentage of lymphocytes, lymphocyte count, percentage of monocytes, white blood cell count, ALT, AST, creatine kinase, neutrophilic granulocyte percentage and hemoglobin were 0–10 mg/L, 20–50%, 1.10–$3.20\times10^9$/L, 3.0–10.0%, 3.5–$9.5\times10^9$/L, 7–40 U/L, 13–35 U/L, 40–200 U/L, 40–75%, 115–150 g/L, respectively.

discharge, 20 (29.9%) had complete resolution with a median interval of 38 (range, 30–53) days between symptom onset and revisit CT after discharge. A total of 340 CT scans were obtained from 106 patients. The average number of CT scans was 3 (range, 1–8).

## CT findings of OP pattern in COVID-19

Of 1285 lesions in 106 patients, pure GGO (32.2%) was the predominant CT finding, followed by the mixed GGO and consolidation (21.2%), mixed GGO, consolidation and linear opacity (17.7%). Pure linear opacity, reversed halo signs and crazy paving were rare, accounting for 1.9%, 2.2% and 2.0% respectively. Mean total CT score was 5.1 ± 2.8. For number of involved lobes, the lesions were mostly located in bilateral lower lobes (right, 24.7%; left, 22.8%) (S1 Table).

Significant differences of pure GGO, pure consolidation, pure linear opacity, mixed consolidation and linear opacity, involvement of lung lobes and total CT score were found among the three time groups (day 0–7, day 8–14, day > 14) (all $P < 0.05$). From day 0–7 to day 8–14, the percentage of pure GGO significantly decreased (41.4% vs. 30.6%, $P = 0.002$); despite no significance with Bonferroni correction, involvement of lung lobes and CT score remarkably increased. From day 8–14 to day > 14, the percentage of pure liner opacity significantly increased (1.0% vs. 3.3%, $P = 0.03$) (Fig 2; S2 Table).

## Comparisons of clinical and CT findings between non-severe and severe cases

Significant differences between non-severe and severe cases were found in terms of age (44.8 ± 13.5 vs. 59.9 ± 12.5 years, $P < 0.001$), presence of comorbidity (27.5% vs. 59.3%, $P = 0.006$), symptom of chest tightness and/or breath shortness (13.9% vs. 40.7%, $P = 0.003$) and decreased lymphocyte count (30.8% vs. 61.5%, $P = 0.005$) (Table 2).

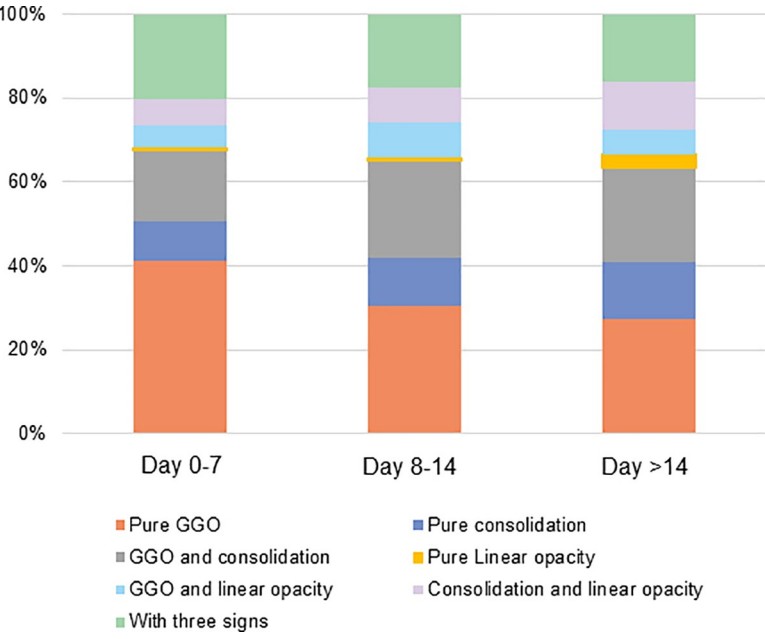

**Fig 2. The evolution of CT findings across the three time groups (day 0–7, day 8–14, day >14) in COVID-19 patients with organizing pneumonia pattern.** GGO = ground glass opacity; with three signs = GGO, consolidation and linear opacity.

**Table 2. Comparisons of clinical characteristics and CT findings between non-severe and severe groups in COVID-19 patients with organizing pneumonia.**

| Characteristics | Non-severe | Severe | P value |
|---|---|---|---|
| | (n = 79) | (n = 27) | |
| **Age (year)[a]** | 44.8 ± 13.5 | 59.9 ± 12.5 | **< 0.001** |
| **Male sex** | 40 (50.6%) | 8 (29.6%) | 0.060 |
| **Comorbidity** | 22 (27.5%) | 16 (59.3%) | **0.006** |
| **Exposure history** | | | |
| Recent travel to Wuhan | 41 (51.9%) | 23 (85.2%) | |
| Contact with infected patient | 27 (34.2%) | 3 (11.1%) | |
| Unknown exposure | 11 (13.9%) | 1 (3.7%) | |
| **Initial symptom** | | | |
| Fever | 68 (86.1%) | 26 (96.3%) | 0.150 |
| Cough | 43 (54.4%) | 19 (70.4%) | 0.150 |
| Expectoration | 17 (21.5%) | 4 (14.8%) | 0.450 |
| Fatigue | 12 (15.2%) | 1 (3.7%) | 0.120 |
| Chest tightness and/or breath shortness | 11 (13.9%) | 11 (40.7%) | **0.003** |
| Pharyngalgia | | | |
| Muscle soreness | 6 (7.6%) | 1 (3.7%) | 0.480 |
| Headache | 6 (7.6%) | 1 (3.7%) | 0.480 |
| Nausea and/or vomiting | 0 (0) | 1 (3.7%) | 0.090 |
| Diarrhea | 1 (1.3%) | 3 (11.1%) | **0.020** |
| No obvious symptoms | 2 (2.5%) | 0 (0) | 0.400 |
| **Laboratory test on admission** | | | |
| C-reactive protein (mg/L) | | | 0.650 |
| 0–10 | 29 (39.7%) | 9 (34.6%) | |
| >10 | 44 (60.3%) | 17 (65.4%) | |
| <0 | 0 (0) | 0 (0) | |
| Percentage of lymphocytes (%) | | | 0.340 |
| 20–50 | 47 (59.5%) | 13 (50.0%) | |
| >50% | 3 (3.8%) | 0 (0) | |
| <20% | 29 (36.7%) | 13 (50%) | |
| Lymphocyte count (×10$^9$/L) | | | **0.005** |
| 1.10–3.20 | 54 (69.2%) | 10 (38.5%) | |
| >3.20 | 0(0) | 0(0) | |
| <1.10 | 24 (30.8%) | 16 (61.5%) | |
| Percentage of monocytes (%) | | | 0.590 |
| 3.0–10.0 | 57 (72.2%) | 18 (72.0%) | |
| >10.0 | 19 (24.1%) | 7 (28.0%) | |
| <3.0 | 3 (3.8%) | 0 (0) | |
| White blood cell count (×10$^9$/L) | | | 0.690 |
| 3.5–9.5 | 54 (68.4%) | 19 (73.1%) | |
| >9.5 | 2 (2.5%) | 0 (0) | |
| <3.5 | 23 (29.1%) | 7 (26.9%) | |
| Alanine Aminotransferase (U/L) | | | 0.39 |
| 7–40 | 63 (80.8%) | 19 (73.1%) | |
| >40 | 13 (16.7%) | 7 (26.9%) | |
| <7 | 2 (2.6%) | 0 (0) | |
| Aspartate Aminotransferase (U/L) | | | 0.180 |
| 13–35 | 61 (78.2%) | 21 (80.8%) | |

(*Continued*)

**Table 2.** (Continued)

| Characteristics | Non-severe | Severe | P value |
|---|---|---|---|
| | (n = 79) | (n = 27) | |
| >35 | 17 (21.8%) | 4 (15.4%) | |
| <13 | 0 (0) | 1 (3.8%) | |
| Creatine kinase (U/L) | | | 0.770 |
| 40–200 | 66 (84.6%) | 18 (78.3%) | |
| >200 | 5 (6.4%) | 2 (8.7%) | |
| <40 | 7 (9.0%) | 3 (13.0%) | |
| Neutrophil percentage (%) | | | 0.910 |
| 40–75 | 52 (65.8%) | 17 (65.4%) | |
| >75 | 19 (24.1%) | 7 (26.9%) | |
| <40 | 8 (10.1%) | 2 (7.7%) | |
| Hemoglobin (g/L) | | | 0.060 |
| 115–150 | 62 (78.5%) | 17 (62.9.0%) | |
| >150 | 6 (7.6%) | 0 (0) | |
| <115 | 11 (13.9%) | 10 (37.1%) | |
| **CT findings** | | | |
| Pure GGO | 320 (31.0%) | 95 (37.4%) | 0.050 |
| GGO and consolidation | 199 (19.3%) | 73 (28.7%) | **0.001** |
| Pure consolidation | 109 (10.6%) | 39 (15.4%) | **0.030** |
| Pure linear opacity | 17 (1.6%) | 7 (2.8%) | 0.240 |
| GGO and linear opacity | 76 (7.4%) | 7 (2.8%) | **0.007** |
| Consolidation and linear opacity | 107 (10.4%) | 9 (3.5%) | **0.001** |
| With three signs | 203 (19.7%) | 24 (9.4%) | < **0.001** |
| **Involvement of lung lobes** | | | < **0.001** |
| Number of affected lobes ≤ 3 | 116 (40.7%) | 6 (10.9%) | |
| Number of affected lobes > 3 | 169 (59.3%) | 49 (89.1%) | |

Note: Unless otherwise indicated, data are reported as the number of patients, with percentages in parentheses.

[a], data were reported as the mean ± standard derivation. Abbreviations: GGO = ground glass opacity; with three signs = GGO, consolidation and linear opacity

There were 1031 and 254 lesions in non-severe and severe cases, respectively. The percentage of mixed GGO and consolidation and pure consolidation were significantly lower in non-severe group than those in severe group (19.3% vs. 28.7%, $P = 0.001$; 10.6% vs. 15.4%, $P = 0.030$), while percentage of mixed GGO and linear opacity, mixed consolidation and linear opacity, mixed three signs were significantly higher in non-severe than severe groups (7.4% vs. 2.8%, $P = 0.007$; 10.4% vs. 3.5%, $P = 0.001$; 19.7% vs. 29.4%, $P < 0.001$). The number of affected lobes > 3 and total CT score were significantly lower in non-severe than severe groups (59.3% vs. 89.1%, $P < 0.001$; 4.7 ± 2.5 vs. 7.5 ± 2.9, $P < 0.001$) (Table 2).

## Comparisons of clinical and CT findings between complete absorption and residual groups

Significant demographics and laboratory test difference between complete absorption and residuals groups were found in age (34.9 ± 9.0 vs. 47.9 ± 13.7, $P = 0.001$), elevated C-reactive protein (31.6% vs. 70.5%, $P = 0.009$) and neutrophil percentage (10.0% vs. 37.0%, $P = 0.020$) (S3 Table).

At day 0–7, 81 and 185 lesions were found on CT in complete absorption and residual groups, respectively. Percentage of pure consolidation was significantly higher in complete absorption than residuals groups (14.8% vs. 5.9%, $P = 0.020$) (Fig 3, S4 Table).

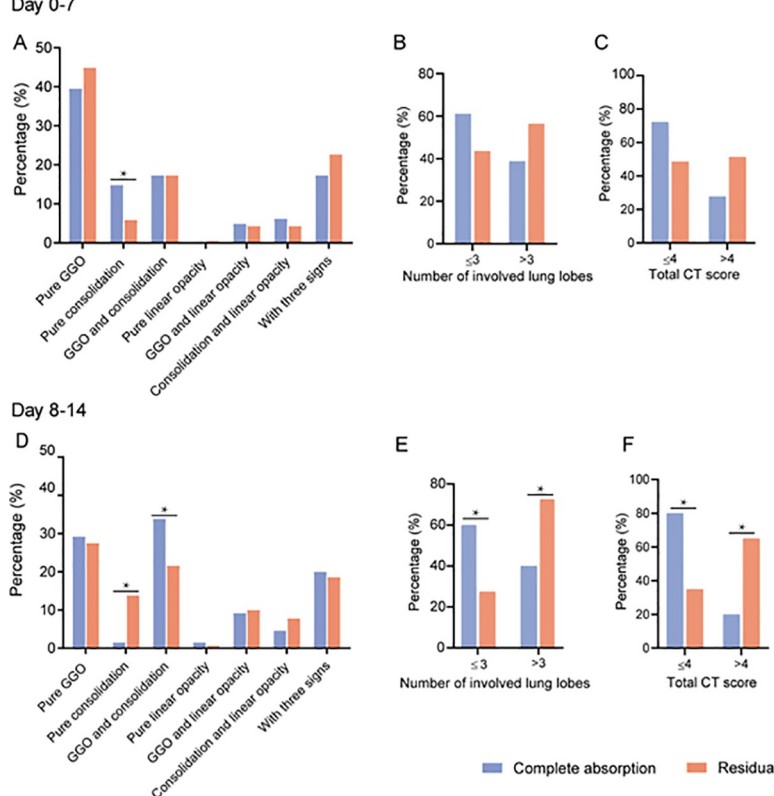

**Fig 3. Comparisons of CT findings between patients with complete absorption and residuals after discharge.**
GGO = ground glass opacity; with three signs = GGO, consolidation and linear opacity. *, $P< 0.05$ indicated
significant differences.

A total of 65 and 269 lesions in complete absorption and residuals groups were found at
day 8–14. Significant differences between complete absorption and residuals groups were
found in percentage of mixed GGO and consolidation (33.8% vs. 21.6%, $P = 0.040$), percentage
of pure consolidation (1.5% vs. 13.8%, $P = 0.010$), percentage of affected lobe number $> 3$
(40.0% vs. 72.5%, $P = 0.03$) (Fig 3, S4 Table). In addition, ROC curve analysis estimated a cut-
off CT score of 4 to discriminate the complete absorption from residuals groups at day 8–14
(AUC = 0.732, $P = 0.006$) (S1 Fig). Significantly higher percentage of CT score $> 4$ was found
in residuals than complete absorption groups (65.0% vs. 20.0%, $P = 0.001$) (Fig 3, S4 Table)
Two cases with series CT scans were presented in Figs 4 and 5.

## Discussion

Regarding the most common OP pattern in COVID-19, this study systematically described the
clinical characteristics and time-dependent evolution of CT findings, as well as later outcome
of patients. Results indicated that 74.5% of OP cases were non-severe and 97.1% cases had
good prognosis with recovery. As for pulmonary resolution, approximately one-third of OP
cases had complete absorption of lesions during day 30–53 after symptom onset while those
with increased percentages of consolidation, number of involved lung lobe $> 3$, and CT
score $> 4$ at week 2 after onset are prone to have pulmonary residuals.

Being consistent with previous finding of COVID-19 [15], the dominant finding in OP pat-
tern was the presence of GGO, followed by mixed GGO and consolidation, with peripheral

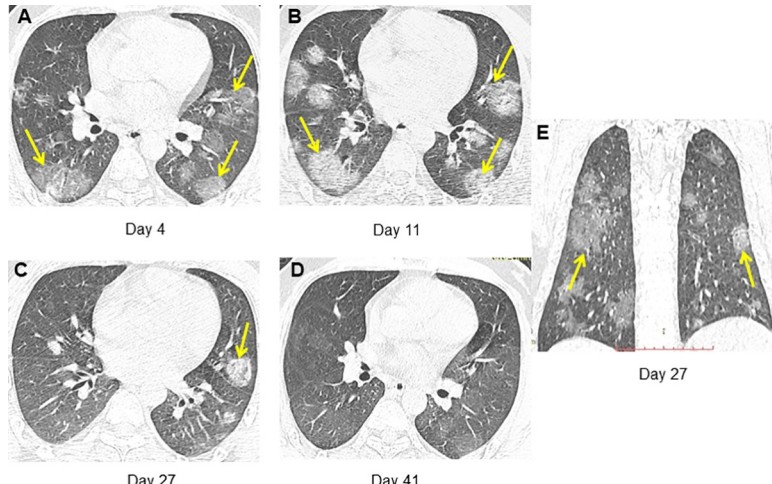

**Fig 4. A 66-year-old man who had been to Wuhan had fever for 4 days and discharged at day 28 after symptom onset.** (A) CT obtained on admission at day 4 after symptom onset shows multiple subpleural consolidation. (B) CT on day 11 shows progression with increased number and size of lung lesions. (C, E) CT on day 27 shows absorption with decreased density of the lesions. (D) CT on day 41 shows almost complete absorption of pulmonary lesions.

and lower lobes distribution. In addition, time-dependent evolution indicated that percentage of GGO decreased while that of mixed GGO and consolidation increased from 1 to 2 weeks after onset, and linear opacity increased from 2 to 3 weeks after onset. These findings were in accordance with prior reports regarding the radiological aggravation ($\leq$ 2 weeks) and improvement ($>$ 2 weeks) in COVID-19 [9, 10]. A systematic review in COVID-19 also observed that GGO turned into extensive consolidation with greatest severity at around day 10

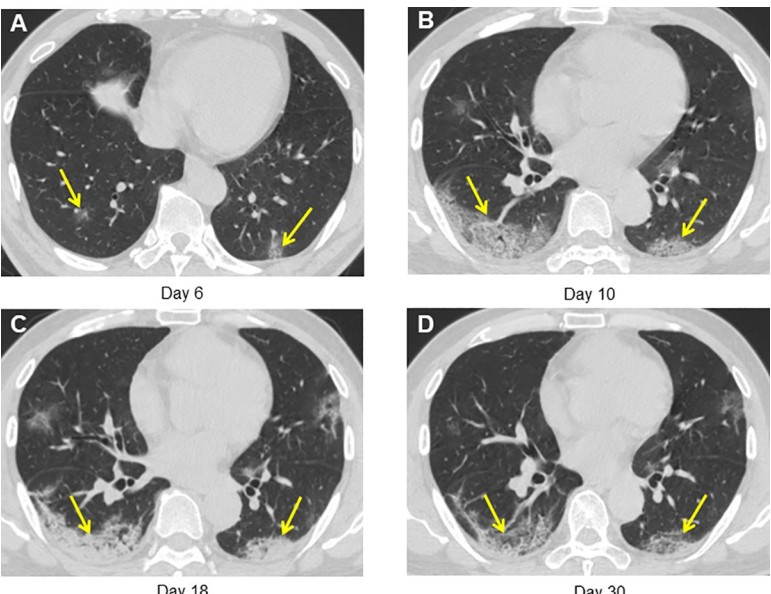

**Fig 5. A 55-year-old man who had fever for 6 days and discharged at day 31 after symptom onset.** (A) CT obtained on admission at day 6 after symptom onset shows focal GGO in bilateral lower lobe. (B) CT on day 10 shows progression with increased size and density of lung lesions. (C) CT on day 18 shows progression with increased density of the lesions. (D) CT on day 30 shows patchy consolidation and linear opacities of pulmonary lesions.

after onset and consolidation gradually resolved after 2 weeks [16]. Histopathologically, alveolar epithelium injured, inflammatory component, fibrin deposition and matrix leaked and coagulated in early and advanced stage and then gradually receded and resorbed during the late disease course [17], which may explain the primary CT findings in OP pattern.

In COVID-19, 74.5% of OP pattern cases were non-severe which consisted with the mild degree of lung injury in most OP cases [18]. In addition, severe cases had older age, more prevalence of comorbidity, and decreased lymphocyte count than non-severe cases. Besides, more prevalence of mixed GGO and consolidation, and pure consolidation with higher lung severity were found in severe cases, whereas non-severe cases showed more prevalence of linear opacity. This may imply a progressive pulmonary involvement in severe cases while a reparative process in non-severe cases. It may be such facts that led to the longer course of disease from symptom onset to discharge in severe than non-severe cases (36.5 ± 9.9 vs. 23.9 ± 7.4 days, $P < 0.001$).

As for clinical outcome, most of OP pattern cases showed good prognosis with discharge. This resembled the previous study of OP [18]. It is noting that for 3 patients with adverse outcomes, they had progressively diffuse GGO and consolidation with interlobular septal thickening in both lungs within the first week after onset. This may be linked to a fast progression from OP to diffuse alveolar damage [19]. Among the discharged patients, those with increased percentages of consolidation, number of involved lung lobe > 3, and CT score > 4 at week 2 after onset were prone to have pulmonary residuals. What we observed during the first two weeks was probably correlated with the underlying organizing process of lung injury [7]. Prior study found that extensive consolidation as well as increased CT score may suggest the disease progression [11, 12]. In this regard, cases with extensive consolidation and progressive lung involvement may have a protracted disease course of lesion absorption. Recently, Zhao et al followed up COVID-19 survivors and found that three quarters of the cohort showed radiological and pulmonary function abnormalities at 3 months after discharge [20]. You et al. discovered that 83.3% of rehabilitating COVID-19 patients still had residual CT abnormalities one month later after discharge, including GGO and pulmonary fibrosis [21]. Similarly, radiological sequelae were also observed in Severe Acute Respiratory Syndrome (SARS) and Middle East Respiratory Syndrome (MERS). In details, radiological sequelae with impaired pulmonary function was found in MERS at the 1-year after infection [22]. Antonio et al. found that some SARS patients showed residual abnormalities on CT with average interval of 18 days after discharge, which was similar to our study [23]. Follow-up studies found that pulmonary parenchymal fibrosis occurred in a substantial portion of SARS-CoV and MERS-CoV patients after discharge [24, 25], which was characterized by GGO, pulmonary fibrosis and pleural thickening [26]. Note that radiological sequelae from SARS and MERS partially accounted for the repaired lung function [23], such as complaint of limitation in general physical function and/or shortness of breath in the early rehabilitation phase [25]. Differently, slighter residuals mainly presenting with linear opacities was found in OP pattern of COVID-19 and the proportion of pulmonary residuals after discharge was lower in our cohort, which may be linked to our cohort coming from non-epidemic areas outside Wuhan and most patients were non-severe. Beyond, elevated C-reactive protein and neutrophil percentage may indicate the state of tissue injury and/or inflammation [27]. Previous study indicated that continuous high levels of C-reactive protein in respiratory infections increases the risk of progression to a critical disease state [28]. In this regard, elevated C-reactive protein and neutrophil percentage may be predictive of radiological sequelae. Although OP cases of COVID-19 had a favourable prognosis, early monitoring and detection of adverse outcomes and radiological sequelae would contribute to the early intervention for those with potential risks to fibrosis, respiratory failure and death [29].

There were several limitations in this study. First was the relatively small sample and retrospective nature. A larger sample is required to further validate the findings regarding OP of COVID-19. Second, the follow-up period for patients is relatively short and many of residual lesions on CT may be reversible, a long-term follow-up in conjunction with lung function tests would help to further clarify the evolution of residual lesions and its relations with lung function. Third, although pathological evidence is scarce, highly resembled CT features of OP were used to define the OP pattern of COVID-19 in this study. Pathological studies are still needed to validate the OP of COVID-19.

In conclusion, as a most common pattern of COVID-19, majority of OP cases were mild or common and had good prognosis. Approximately one-third of OP cases had complete absorption of lesions during 1–2 month after symptom onset while those with increased frequency of pure consolidation, number of involved lung lobe > 3, and CT score > 4 at week 2 after symptom onset were prone to have pulmonary residuals.

## Supporting information

**S1 Table. CT findings in COVID-19 patients with organizing pneumonia pattern.** Abbreviations: GGO = ground glass opacity; with three signs = GGO, consolidation and linear opacity.
(DOCX)

**S2 Table. CT findings during different time groups in COVID-19 patients with organizing pneumonia pattern.** Note: *Significance at $P<0.017$ with Bonferroni correction. Abbreviations: GGO = ground glass opacity; with three signs = GGO, consolidation and linear opacity.
(DOCX)

**S3 Table. Comparisons of demographics and laboratory test between complete absorption and residual groups in COVID-19 patients with organizing pneumonia.** Note: Unless otherwise indicated, data are reported as the number of patients, with percentages in parentheses. [a], data were reported as the mean ± standard derivation.
(DOCX)

**S4 Table. Comparisons of CT findings between complete absorption and residuals groups.** Abbreviations: GGO = ground glass opacity; with three signs = GGO, consolidation and linear opacity.
(DOCX)

**S1 Fig. ROC curve for total CT score to distinguish residuals group from complete absorption group at day 8–14.**
(DOCX)

**S1 File.**
(XLSX)

## Author Contributions

**Conceptualization:** Chao Jin, Carol C. Wu, Zhe Liu.

**Data curation:** Yan Wang, Huifang Zhao, Ting Liang, Zhe Liu, Zhijie Jian, Runqing Li, Zekun Wang, Fen Li, Jie Zhou, Shubo Cai, Yang Liu, Hao Li, Yukun Liang.

**Formal analysis:** Yan Wang, Cong Tian.

**Investigation:** Yan Wang, Fen Li, Jie Zhou, Yang Liu, Hao Li.

**Methodology:** Yan Wang, Chao Jin, Huifang Zhao, Ting Liang, Zhe Liu, Zhijie Jian, Runqing Li, Yang Liu, Jian Yang.

**Project administration:** Yan Wang, Huifang Zhao, Yukun Liang, Jian Yang.

**Resources:** Huifang Zhao, Fen Li, Jie Zhou, Shubo Cai, Jian Yang.

**Supervision:** Chao Jin, Jian Yang.

**Writing – original draft:** Yan Wang, Chao Jin.

**Writing – review & editing:** Cong Tian, Jian Yang.

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
