## [Decision Letter · Decision Letter 0]

24 Jul 2020

PONE-D-20-17536

Organizing Pneumonia of COVID-19: Time-dependent Evolution and Outcome in CT Findings

PLOS ONE

Dear Dr. Yang,

Thank you for submitting your manuscript to PLOS ONE. After careful consideration, we feel that it has merit but does not fully meet PLOS ONE’s publication criteria as it currently stands. Therefore, we invite you to submit a revised version of the manuscript that addresses the points raised during the review process.

We look forward to receiving your revised manuscript.

Kind regards,

Wenbin Tan

Academic Editor

PLOS ONE

Journal Requirements:

3.

In your Data Availability statement, you have not specified where the minimal data set underlying the results described in your manuscript can be found. PLOS defines a study's minimal data set as the underlying data used to reach the conclusions drawn in the manuscript and any additional data required to replicate the reported study findings in their entirety. All PLOS journals require that the minimal data set be made fully available. For more information about our data policy, please see http://journals.plos.org/plosone/s/data-availability.

Editor Comments:

This manuscript reports dynamics of lung injuries and evolution of CT findings in COVID-19 patients. They found 1/3 of cases had complete absorption of lesions in 1-2 months; while many other patients (47 out of 67) still showed lesional residuals which may indicate a long-term pulmonary sequelae. It is a very well presented manuscript and the observational data supports the conclusion well. However, there are some minor issues for the authors to consider.

(1) the format of number, such as "79(74.5%) patients were non-severe and 103(97.2%)", a space should be inserted between "number" and "(..%)". Please change throughout the entire manuscript.

(2) IRB protocol of XJTU1AF2020LSK-011 is a multiple-center single IRB? This needs to be confirmed and clarified in the content since patients from multiple hospitals being involved. However, the attached copy of IRB approval letter is lack of such crucial information which is required.

(3) table 2&3, the laboratory test results on admission should be presented as "median (IQR)"

(4) footnote for table 3, "with three signs = GGO, consolidation and linear opacity", please fix the typos.

(5) Table 3 can be deleted, a description of significant results in the content should be sufficient.

(6) Fig 4, using arrows in the images to indicate described lesions in the legend will be helpful.

(7) in discussion, what will be potential long-term pulmonary sequelae in COVID-19? Authors should cite more available evidence regarding long-term pulmonary sequelae in SARS and MERS patients to support their prediction, particularly lung fibrosis.

(8) clarify "characteristics unmatched with OP pattern" in the method.

Reviewers' comments:

Reviewer's Responses to Questions

**Comments to the Author**

1. Is the manuscript technically sound, and do the data support the conclusions?

Reviewer #1: Yes

2. Has the statistical analysis been performed appropriately and rigorously? 

Reviewer #1: Yes

3. Have the authors made all data underlying the findings in their manuscript fully available?

Reviewer #1: Yes

4. Is the manuscript presented in an intelligible fashion and written in standard English?

Reviewer #1: Yes

5. Review Comments to the Author

Reviewer #1: Overall a good and well written manuscript.

Please grammar check the manuscript.

Title:Ok

Abstarct:Ok

Introduction and Aim: Ok

Materials and methods : Were there any cases of mixed infections?

Results:Ok

Discussion: Ok

Conclusion: Ok

Tabels : Ok

Figures:Please add few more CT images showing evolution of the findings.

6. PLOS authors have the option to publish the peer review history of their article (what does this mean?). If published, this will include your full peer review and any attached files.

Reviewer #1: No

---

## [Author Response · Author response to Decision Letter 0]

14 Sep 2020

Editor Comments:

(1) the format of number, such as "79(74.5%) patients were non-severe and 103(97.2%)", a space should be inserted between "number" and "(..%)". Please change throughout the entire manuscript.

Re to ED1: We are so appreciated for your warm help and valuable comments in reviewing our paper that indeed make us think deeply and improve our work.

According to your suggestion, we have revised the numerical format of the entire manuscript. In addition, we have checked the whole text and revised the related details.

(2) IRB protocol of XJTU1AF2020LSK-011 is a multiple-center single IRB? This needs to be confirmed and clarified in the content since patients from multiple hospitals being involved. However, the attached copy of IRB approval letter is lack of such crucial information which is required.

Re to ED2: Thanks for your comments. This multicenter study was launched by the institution of the first author, and approved by all the multicenter institutions. For ethic issue, IRB protocol of XJTU1AF2020LSK-011 is a multiple-center and we have clarified this in the attached copy of IRB approval letter.

(3) table 2&3, the on admission should be presented as "median (IQR)"

Re to ED3: Thanks for your comments. At the initial of this multicenter study, we used a general protocol for data collection. The categorized format of laboratory test (increase, decrease or within the normal range) was utilized. Given this, the data with count (percentage) was calculated and reported in the results.

(4) footnote for table 3, "with three signs = GGO, consolidation and linear opacity", please fix the typos.

Re to ED4: Thanks for your comments. The typos of “with three signs = GGO, consolidation and linear opacity” have been revised in the paper. 

(5) Table 3 can be deleted, a description of significant results in the content should be sufficient.

Re to ED5: Thanks for your suggestion. The related contents have been revised as “Significant demographics and laboratory test difference between complete absorption and residuals groups were found in age (34.9 ± 9.0 vs. 47.9 ± 13.7, P = 0.001), elevated C-reactive protein (31.6% vs. 70.5%, P = 0.009) and neutrophil percentage (10.0% vs. 37.0%, P = 0.020)”. And Table 3 was placed in the supplementary materials to provide a more detailed reference to readers.

(6) Fig 4, using arrows in the images to indicate described lesions in the legend will be helpful. 

Re to ED6: Thanks for your suggestion. The arrows to indicate described lesions in the legend have been added in the figure 4.

(7) in discussion, what will be potential long-term pulmonary sequelae in COVID-19? Authors should cite more available evidence regarding long-term pulmonary sequelae in SARS and MERS patients to support their prediction, particularly lung fibrosis.

Re to ED7: Thanks for your comments. Recent studies investigated pulmonary sequelae of COVID-19 and discovered that a substantial number of patients had radiological abnormalities after discharge, including GGO and pulmonary fibrosis [1, 2]. However, the impact of long-term pulmonary sequelae of COVID-19 on lung function remains to be validated in future. Previous follow-up chest radiographs obtained from the patients with MERS-CoV and SARS-CoV showed a substantial portion (around 30%) chest radiographic abnormalities, which were characterized by the presence of pulmonary fibrosis, GGO, and pleural thickening [3-5]. Note that radiological sequelae from SARS and MERS partially accounted for the repaired lung function, such as complaint of limitation in general physical function and/or shortness of breath in the early rehabilitation phase. Differently, slighter residuals mainly presenting with linear opacities was found in OP pattern of COVID-19 and the proportion of pulmonary residuals after discharge was lower in our cohort, which may be linked to our cohort coming from non-epidemic areas outside Wuhan and most patients were non-severe. We have thoroughly discussed this in discussion part.

(8) clarify "characteristics unmatched with OP pattern" in the method.

Re to ED8: Thanks for your precious comments. Referring to prior articles of H1N1 pneumonia [6], CT findings may simulate organizing pneumonia (OP), bronchopneumonia, or acute interstitial pneumonia (AIP) pattern. In detail, the lesions were considered to belong to OP pattern when lung abnormalities showed areas of consolidation or GGO showing lower lung zone predominance and being distributed along the subpleural lungs or bronchovascular bundles. They were regarded to belong to AIP pattern when the abnormalities demonstrated areas of consolidation with or without GGO showing patchy and extensive distribution without zonal predominance. And they were presumed to belong to bronchopneumonia pattern when the abnormalities had combination of consolidation, GGO, small centrilobular nodules and bronchial or bronchiolar wall thickening. We have clarified the characteristics unmatched with OP pattern as “The pulmonary lesions were considered to unmatched with OP pattern: (1) extensive GGO and/or consolidation diffusely in both lungs; (2) total CT score greater than 10; (3) combined consolidation and GGO with bronchial or bronchiolar wall thickening. Patients which were with unmatched with OP pattern or unqualified CT images were excluded” in the materials and methods part.

References

1. Zhao Y, Shang Y, Song W, Li Q, Xie H, Xu Q, et al. Follow-up study of the pulmonary function and related physiological characteristics of COVID-19 survivors three months after recovery. EClinicalMedicine 2020:100463.

2. You J, Zhang L, Ni-jia-ti M, Zhang J, Hu F, Chen L, et al. Anormal pulmonary function and residual CT abnormalities in rehabilitating COVID-19 patients after discharge. J Infect. 2020, 16:32. doi:10.1016/j.jinf.2020.06.003.

3. Xie L, Liu Y, Fan B, Xiao Y, Tian Q, Chen L, et al. Dynamic changes of serum SARS-coronavirus IgG, pulmonary function and radiography in patients recovering from SARS after hospital discharge. Respir Res. 2005 Jan 8;6(1):5.

4. Chan KS, Zheng JP, Mok YW, Li YM, Liu Y, Chu C, et al. SARS: prognosis, outcome, and sequelae. Respirology. 2003 Nov;8 Suppl (Suppl 1):S36-40.

5. Das K, Lee E, Singh R, Enani M, Dossari K, Gorkom K, et al. Follow-up chest radiographic findings in patients with MERS-CoV after recovery. Indian J Radiol Imaging. 2017;27: 342. doi:10.4103/ijri.IJRI_469_16. 

6. Kang H, Lee KS, Jeong YJ, Lee HY, Kim K Il, Nam KJ. Computed tomography findings of influenza a (H1N1) pneumonia in adults: Pattern analysis and prognostic comparisons. J Comput Assist Tomogr. 2012. doi:10.1097/RCT.0b013e31825588e6.

Reviewer #1’s Comments:

(1) Materials and methods: Were there any cases of mixed infections?

Re to RE1: We appreciate very much for your kindly help and valuable comments in reviewing our paper that indeed make us think deeply and improve our work.

Recent studies have reported co-infections in people with COVID-19. According to recent paper [1], Lansbury et al. conducted a systematic review and meta-analysis of patients with coinfections in COVID-19 and they found 7% of hospitalized COVID-19 patients had a bacterial co-infection and 3% had viral co-infection. The commonest bacteria were Mycoplasma pneumonia, Pseudomonas aeruginosa and Haemophilus influenzae while the commonest virus were Respiratory Syncytial Virus and influenza A. However, in our cohort, no cases had mixed infections, this may be because that most of our patients were from non-epidemic areas outside Wuhan, and most patients were non-severe. We have clarified this in results part.

References

1. Lansbury L, Lim B, Baskaran V, Lim WS. Co-infections in people with COVID-19: a systematic review and meta-analysis. J Infect. 2020;81: 266–275. doi:https://doi.org/10.1016/j.jinf.2020.05.046.

(2) Figures: Please add few more CT images showing evolution of the findings.

Re to RE1: Thanks for your suggestion. We have added other CT images as figure 5 to present evolution of findings. This was a 55-year-old male who had fever for 6 days and discharged 1 month later. CT obtained on day 6 after admission showed focal GGO on bilateral lower lobes. On day 10 CT showed progressed mixed consolidation and on day 18 CT showed mixed consolidation and linear opacities. The day before discharge the patient performed CT examination and found that lesions were partially absorbed and there were residual consolidation and linear opacities in bilateral lower lobes.

Response to Editor’s Comments

Re to ED1: Thanks for your comments. We have revised the current ethics statement and included the full name of the ethics committee/institutional review board(s) as following: “ This multicenter study was launched by the First Affiliated Hospital of Xi’an Jiaotong University, and approved by all the multicenter institutions, including the First Affiliated Hospital of Xi’an Jiaotong University, the Eighth Hospital of Xi’an, Xi’an Chest Hospital, Ankang Center Hospital, Wuhan No.9 Hospital, Hanzhong Center Hospital, Baoji Center Hospital. For ethic issue, IRB protocol of XJTU1AF2020LSK-011 is multiple-center.” The above statement has been added at the beginning of the Methods section of our manuscript file.

2) We note that your Data Availability Statement reads, "Since this is multi-center cooperation data, we do not have full access to it, so we can only share part of the data from our hospital. All relevant data are within the manuscript and its Supporting Information files."

Could you please clarify whether "Supporting information.zip" contains your full Minimal Data Set, or whether the data from additional centers/hospitals is necessary for replication?

Re to ED2: Thanks for your suggestion. Due to the data privacy protection policy, we have tried our best to communicate with multi-center ethics review board and sorted minimal data set to support our conclusion. This minimal data set was from multiple hospitals and was able to replicate our conclusions.

---

## [Editor Report · Decision Letter 1]

25 Sep 2020

Organizing Pneumonia of COVID-19: Time-dependent Evolution and Outcome in CT Findings

PONE-D-20-17536R1

Dear Dr. Yang,

We’re pleased to inform you that your manuscript has been judged scientifically suitable for publication and will be formally accepted for publication once it meets all outstanding technical requirements.

Kind regards,

Wenbin Tan

Academic Editor

PLOS ONE

Additional Editor Comments: The revised manuscript has addressed the comments well. This is a very important report to show the potential sequelae in discharged COVID-19 patients, though 3-month follow-up time is bit short as a limitation. I would look forward to a year or even longer time follow-up study from the same group in future.

---

## [Editor Report · Acceptance letter]

29 Sep 2020

PONE-D-20-17536R1 

Organizing Pneumonia of COVID-19: Time-dependent Evolution and Outcome in CT Findings 

Dear Dr. Yang:

I'm pleased to inform you that your manuscript has been deemed suitable for publication in PLOS ONE. Congratulations! Your manuscript is now with our production department. 

Kind regards, 

on behalf of

Dr. Wenbin Tan 

Academic Editor

PLOS ONE